# Discriminatively Matched Part Tokens for Pointly Supervised Instance Segmentation

## Abstract

The self-attention mechanism of vision transformer has demonstrated potential for instance segmentation even using a single point as supervision. However, when it comes to objects with significant deformation and variations in appearance, this attention mechanism encounters a challenge of semantic variation among object parts. In this study, we propose discriminatively matched part tokens (DMPT), to extend the capacity of self-attention for pointly supervised instance segmentation. DMPT first allocates a token for each object part by finding a semantic extreme point, and then introduces part classifiers with deformable constraint to re-estimate part tokens which are utilized to guide and enhance the fine-grained localization capability of the self-attention mechanism. Through iterative optimization, DMPT matches the most discriminative part tokens which facilitate capturing fine-grained semantics and activating full object extent. Extensive experiments on PASCAL VOC and MS-COCO segmentation datasets show that DMPT respectively outperforms the state-of-the-art method by 2.0% $mAP_{50}$ and 1.6% AP. DMPT is combined with the Segment Anything Model (SAM), demonstrating the great potential to reform point prompt learning. Code is enclosed in the supplementary material.

## 1 Introduction

In the past few years, the self-attention mechanism (Vaswani et al., 2017) of vision transformers (ViTs) (Dosovitskiy et al., 2021; Touvron et al., 2021; Liu et al., 2021) has achieved success in object localization (Zhou et al., 2016). Due to its ability to establish spatial dependencies among features, the self-attention mechanism has also been widely applied in weakly supervised object localization scenarios (Gao et al., 2021). Nevertheless, in fine-grained segmentation tasks, the attention mechanism remains challenged by the significant semantic variation among object parts, Fig. 1(b) and (c). This issue can be mitigated by providing precise mask annotations (Lin et al., 2014; Everingham et al., 2010; Hariharan et al., 2011), but it entails a substantial human effort for data annotation.

Given the powerful spatial localization potential of the self-attention mechanism, how can we harness it to achieve accurate instance segmentation in scenarios where only point supervision is available, Fig. 1(a)? In tackling this problem, we conducted analysis of two key factors that influence the self-attention maps for instance segmentation: the architecture of self-attention itself, and the guidance in self-attention-based networks. Modifying the self-attention architecture renders existing pre-trained models inapplicable. We adopt the latter approach, where we propose to split each object to parts and use these parts as guidance to steer the self-attention mechanism towards the activation of finer-grained semantics, and thereby achieving more accurate instance segmentation under the point supervision setting. In order to obtain the object parts while ensuring their ability to guide the self-attention mechanism, we encounter two challenges: (1) How to partition an object with deformation to semantic-consistent parts using the coarse attention map generated through point supervision; (2) How to guarantee the same parts of different objects semantically consistent.

In this paper, we propose Discriminatively Matched Part Tokens (DMPT), which models each instance as a set of deformable parts. Such deformable parts are initialized on the attention map generated on a supervision point and optimally and iteratively matched with part classifiers. Using ViT as the backbone network, DMPT performs the following three procedures. **(i) Part token allocation**. Upon the self-attention map, the mean-shift method is carried out to generate part clusters and localize

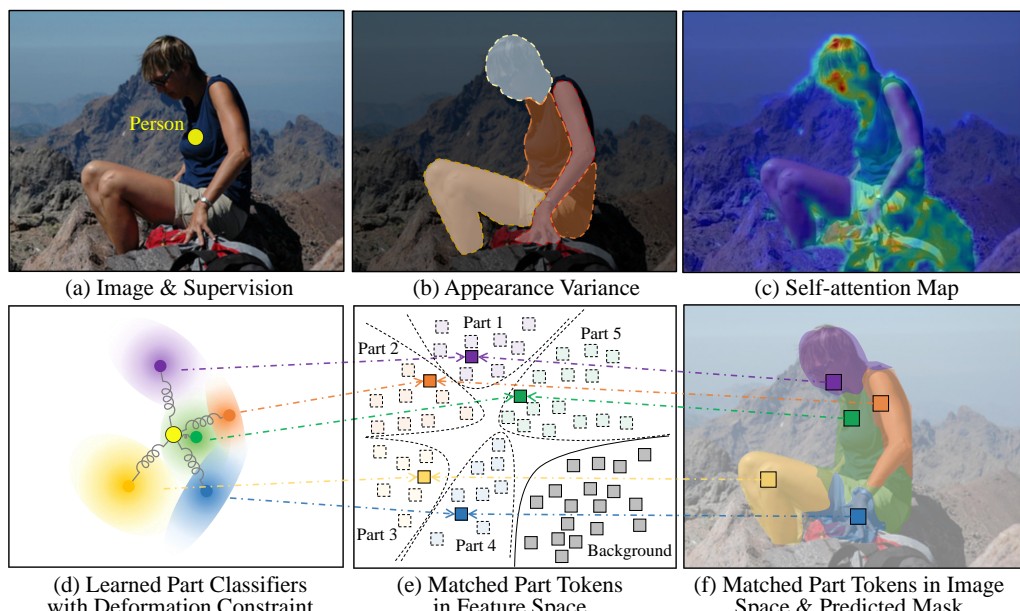

(a) Image & Supervision  (b) Appearance Variance  (c) Self-attention Map

(d) Learned Part Classifiers with Deformation Constraint  (e) Matched Part Tokens in Feature Space  (f) Matched Part Tokens in Image Space & Predicted Mask

Figure 1: Illustration of challenges (upper) of pointly supervised instance segmentation and the proposed approach (lower). **Upper:** The self-attention map produced by ViT ignores object parts with semantic variance when using a single point as supervision. **Lower:** Our DMPT learns multiple part classifiers with deformation constraint (colored ovals) to match the part tokens (colored rectangles), and handle part semantic variance. (Best viewed in color)

semantic extreme points, Fig. 1(d). The part tokens are then initialized using the patch tokens near these semantic extreme points. **(ii) Token-classifier matching.** A set of part classifiers are firstly trained based on the part tokens to capture stable fine-grained semantics by optimizing an instance classification loss, Fig 1(f). As the part annotations are unavailable during training, we introduce a token-classifier matching mechanism under the constraint of the instance classification loss, where each part classifier takes the matched part tokens as input to avoid the semantic aliasing among part classifiers. Considering that the initialized part tokens maybe inaccurate caused by the deformation of objects, we treat the semantic extreme points as "anchors", and used them to define spatial offset constraint for part tokens during the token-classifier matching. **(iii) Part-based guidance generation.** Using these part tokens, a set of part points is generated and utilized as guidance for the self-attention-based network to improve the fine-grained localization capability of the self-attention mechanism in a point-supervised manner. Through this procedure, DMPT extends the capacity of self-attention so that it can handle large deformation and semantic variance.

When combined with Segment Anything Model (SAM), DMPT-SAM improves the instance segmentation performance of vanilla SAM by a significant margin with a single point as prompt, Table. 1, demonstrating its ability to reform point prompt learning.

The contributions of this paper are summarized as follows:

- We propose discriminatively match part tokens (DMPT), extends the capacity of self-attention for pointly supervised instance segmentation (PSIS), providing a systemic way to address large deformation and appearance variance.

- We design simple-yet-effective modules to allocate and optimize part tokens using semantic extreme points and token-classifier matching.

- DMPT achieves the best performance for PSIS, demonstrating the potential to reform point prompt learning.

| Method | PASCAL VOC 2012 | | | MS-COCO 2017 | | |
|---|---|---|---|---|---|---|
| | mAP$_{25}$ | mAP$_{50}$ | mAP$_{75}$ | AP | AP50 | AP75 |
| SAM (Kirillov et al., 2023) | 59.4 | 39.9 | 19.0 | 19.5 | 36.8 | 18.8 |
| DMPT-SAM(ours) | 70.7(+11.3) | 59.4(+19.5) | 35.5(+16.5) | 22.6(+3.1) | 45.7(+8.9) | 21.4(+2.6) |

Table 1: Comparison of DMPT-SAM with vanilla SAM, where ViT-base is used as the backbone.

## 2 RELATED WORK

**Deformable Part-based Models.** Deformable part-based models, $i.e.$, deformable template models(Coughlan et al., 2000; Cootes et al., 2001) and manifold part-based models (Fischler & Elschlager, 1973; Felzenszwalb & Huttenlocher, 2005; Amit & Trouvé, 2007; Burl et al., 1998), achieved great success to handle object appearance variance. In particular, pictorial structure models (Felzenszwalb & Huttenlocher, 2005; Fischler & Elschlager, 1973) captured the geometric arrangement of parts through a set of "springs" that connect pairs of parts. DPM (Felzenszwalb et al., 2010) defined deformation cost to punish parts far away from the root position. DCN (Dai et al., 2017) augmented convolutional kernels by learning offsets of spatial locations to cover irregular object layout. Deformable DETR (Zhu et al., 2021) implemented this idea into vision transformer to further improve the feature representative. PST (Yang et al., 2022) decomposed object to parts by splitting feature vectors belonging to a whole object to multiple subsets using an Expectation-Maximization algorithm. In this study, we are inspired by the conventional DPM (Felzenszwalb et al., 2010) and intend to exploit its potential to enhance the self-attention mechanism of ViTs.

**Weakly Supervised Instance Segmentation.** This pursues segmenting instances given image-level labels as supervision signals. Early researches (RR et al., 2013) segmented the instances from selected proposals using activation maps (Ge et al., 2019). For example, PRM (Zhou et al., 2018) produced a peak response map to select proper proposals. Class activation map (CAM) (Zhou et al., 2016) locate objects by mapping the class score back to the previous convolution layer. BESTIE (Kim et al., 2022) transferred instance heat-maps, $e.g.$ center and offset maps, from weakly supervised segmentation results and refined the maps for accurate segmentation. This line of methods experience difficulty to represent diverse semantics of object parts.

**Pointly Supervised Instance Segmentation.** This task predicts an instance mask for each object using a single point (within object extent) as supervision. Compared to weakly supervised methods, pointly supervised ones provide coarse instance location prompt, while only increasing annotation cost by about 10% (Chen et al., 2022). Recently, PSPS (Fan et al., 2022) generated pseudo mask labels by minimizing traversing distance between each pair of pixel and point label. Point2Mask (Li et al., 2023) improved this idea and proposed a transport cost upon ground-truth points and both of high-level semantics and low-level boundaries, which achieves the SOTA performance on panoptic segmentation. Nevertheless, a single point can be placed on any object parts, which correspond to diverse semantics. Training a model using such diverse semantics leads poor instance segmentation performance. While AttnShift (Liao et al., 2023) estimates fine-grained semantics using a clustering method, it experiences difficulty to elaborate stable part semantics across objects.

Using a large number of mask labels as supervision and point labels as prompt, SAM (Kirillov et al., 2023) greatly improved the generalization capacity of transformer segmentation models. However, SAM lacks a mechanism to handle various object parts, which causes the challenging over-segmentation and/or under-segmentation issues. By defining a part-based ViT model, this study has the potential to be integrated with SAM to enhance the capacity of segmenting object parts.

## 3 METHODOLOGY

### 3.1 PRELIMINARY

Each input image is split to $W \times H$ patch tokens $\mathbf{M} = \{\mu_{i,j} \in \mathbb{R}^{1 \times D}, i = 1, 2, ..., W, j = 1, 2, ..., H\}$, where $D$ is the feature dimension of tokens. To activate object of interests, a set of query tokens are randomly initialized and concatenated with the patch tokens, Fig. 2. Features of query tokens are extracted by the cascaded self-attention mechanism in ViTs, which enables query tokens

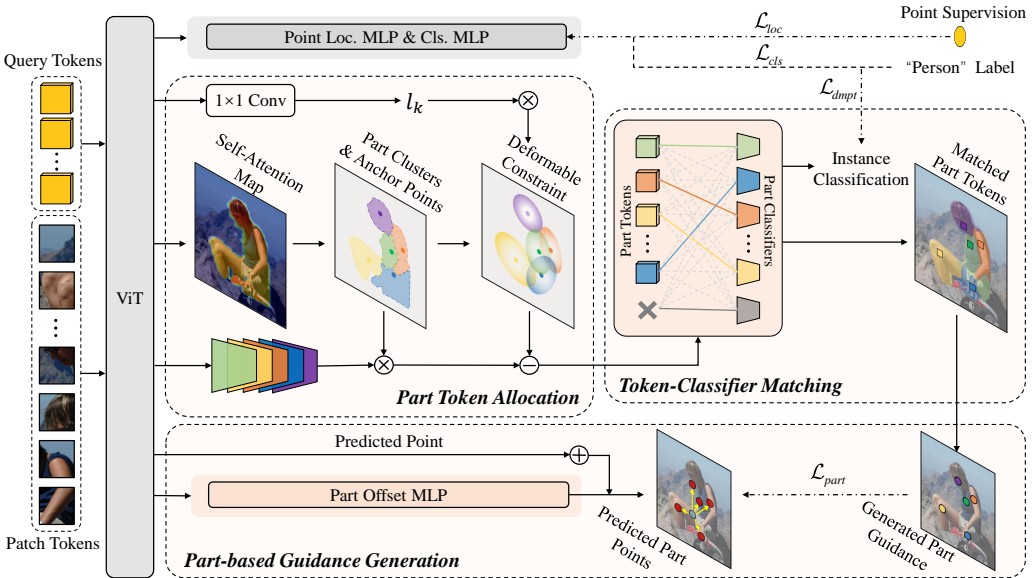

Figure 2: Diagram of DMPT, which utilizes the self-attention map generated by ViT to allocate part tokens with deformation constraint and matches them with the part classifiers to learn fine-grained part semantics. (Best viewed in color)

capturing feature dependency across all patch tokens. Following the self-attention operation (Abnar & Zuidema, 2020; Gao et al., 2021; Liao et al., 2023), we multiply the self-attention maps from shallow to the deep layers to produce a self-attention map $A \in \mathbb{R}^{W \times H}$ for each query token, and $A_{i,j}$ denotes the attention value between the patch token $\mu_{i,j}$ and the query token. [1]

To activate objects, query tokens are respectively passed through two multi-layer perception (MLP) branches to predict a point with class probabilities and coordinates, Fig. 2. The predicted points are assigned to the supervision points or background using a bipartite matching loss

$$\mathcal{L}_{obj} = \mathcal{L}_{loc} + \mathcal{L}_{cls}, \tag{1}$$

where $\mathcal{L}_{loc}$ is the L1-norm loss (Carion et al., 2020) defined upon coordinates of the predicted point and the supervision point. $\mathcal{L}_{cls}$ is the focal loss (Lin et al., 2017) defined upon the point classification probability and the category label. This loss restraints each object can only be assigned to one query token. According to the assigning results, we can obtain the self-attention $A$ for each object/query. Considering that $A$ is a coarse activation map that suffers from background noise and/or object part missing, Fig 1(c). We propose to decompose each object into multiple parts for fine-grained segmentation.

## 3.2 PART TOKEN ALLOCATION

Each object part is expected to be represented with an optimal part token. To this end, the mean-shift method (Comaniciu & Meer, 2002) is employed to group object parts to clusters within the feature space, Fig. 1(e). Let $\mathbf{P} = \{\mathcal{P}_k \in \mathbb{R}^{1 \times D}, k = 1, ..., K\}$ denotes the part tokens, and $\mathbf{C} = \{\mathcal{C}_k \in \mathbb{R}^{1 \times D}, k = 1, ..., K\}$ the cluster centers. Each cluster center $\mathcal{C}_k$ is computed by averaging the features of patch tokens belonging to the cluster. The $k$-th part token $\mathcal{P}_k$ is estimated as

$$\mathcal{P}_k = \arg\max_{\mu_{i,j} \in M^+} \sigma(\mu_{i,j}, \mathcal{C}_k). \tag{2}$$

where $M^+$ indicates patch tokens within the foreground region of attention map $A$. Note that we determine foreground region by a empirical threshold on confidence of $A$. And $\sigma(a, b)$ is for calculating cosine similarity of vector $a$ and $b$.

---

[1]Please refer to the supplementary materials for more details.

Due to object deformation, object parts of the same categories from different images might be falsely grouped. This leads to a certain deviation between representing semantics of object parts with cluster centers and their actual counterparts. To solve this issue, we introduce a well-defined part deformation constraint to optimize part tokens.

**Part Deformation Constraint.** This is defined on the assumption that patch tokens close to cluster centers are more likely to be part tokens. Denote the coordinates of part tokens (cluster centers) estimated by Equ. 2 as $\mathbf{q} = \{q_k = (x_k, y_k), k = 1, 2, ..., K\}$, where $\{x_i = 1, 2, ..., W, y_i = 1, 2, ..., H\}$. $\mathbf{q}$ are termed as anchor points, according to which we define the part deformation constraint. Assume the newly estimated part token $\mathcal{P}_k$ is located at $(x_i, y_i)$ within the $k$-th cluster, its deformation constraint is defined as

$$d_k(x_i, y_i) = l_k \cdot \Delta(q_k, (x_i, y_i)), \tag{3}$$

where deformation features $\Delta(q_k, (x_i, y_i)) \in \mathbb{R}^{1 \times 4}$ is defined as

$$\Delta(q_k, (x_i, y_i)) = (dx_k, dy_k, dx_k^2, dy_k^2) = \left( |x_k - x_i|, |y_k - y_i|, |x_k - x_i|^2, |y_k - y_i|^2 \right), \tag{4}$$

where $l_k$ represents learnable parameters output by an $1 \times 1$ convolutional layer, Fig. 2. We initialize $l_k = (0, 0, 1, 1)$ to represent the squared distance between the location of a part token and its anchor position. Part deformation indicates that the token far from its anchor has a lower probability of being an object part, and vice versa.

To learn stable part semantics across objects, the model should not only leverage clustering mechanisms to explore semantic extremes but also learns a discriminative part semantic model, *i.e.*, part classifiers (introduced in the next subsection). Given the part classification score $\mathbf{s}_k = \{s_k(x_i, y_i) \in [0, 1]\}$ for each patch token $\mu_{x_i, y_i}$ in the $k$-th cluster, the part token $\mathcal{P}_k$ in Equ. 2 is updated as

$$\mathcal{P}_k = \underset{\mu_{x_i, y_i} \in M_{\mathcal{C}_k}}{\arg\max} \ (s_k(x_i, y_i) - \alpha \hat{d}_k(x_i, y_i)), \tag{5}$$

where $M_{\mathcal{C}_k}$ denotes the set of patch tokens belonging to the $k$-th cluster. $\alpha$ is an experimentally determined factor, and $\hat{d}_k(x_i, y_i) \in \mathbb{R}^{1 \times 1}$ the summation of displacements in $d_k(x_i, y_i)$. Equ. 5 indicates that part token allocation essentially seeks a balance between the part classification scores and part deformation constraint, controlled by learnable parameters $l_k$. It guarantees that the allocated part token has a high classification score while being close to the anchor point.

### 3.3 TOKEN-CLASSIFIER MATCHING

Note that Equ. 5 is defined upon the part classification score $\mathbf{s}$ for patch tokens. Nevertheless, $\mathbf{s}$ is unavailable during training as there is only a single point supervision for each instance. Using the attention map $A$ as $\mathbf{s}$ is a possible way, but the appearance variation issue remains.

**Matching with Constraint.** We introduce a set of part classifiers $\mathbf{f} = \{f_n, n = 1, 2, ..., N\}$, where $N$ is the number of parts for an object. Each part classifier is implemented by a single fully connected layer, Fig. 2. $\mathbf{s}_n$ is then predicted by the $n$-th part classifier as $s_n(x_i, y_i) = f_n(\mu_{x_i, y_i})$. Assume that the part classification score of $n$-th part token $\mathcal{P}_n$ is estimated by $n$-th part classifier $f_n$. To learn the part classifiers, we define an instance classification loss for each point supervision as

$$\mathcal{L}_{dmpt} = \text{CE}\left( \sum_{n=1}^{N} \left( f_n(\mathcal{P}_n) - \alpha \hat{d}_n(x_n, y_n) \right), Y \right), \tag{6}$$

where $Y$ is the ground-truth label of the object and $\text{CE}(\cdot)$ the cross-entropy loss (Deng et al., 2009).

In fact, the pair-wise relation between part classifier $f_n$ and part token $\mathcal{P}_k$ is unknown. To construct the relation, a token-classifier matching indicator is added to the instance classification loss. Let $\mathbf{m} = \{m_{n,k}, n = 1, 2, ...N, k = 1, 2, ...K\}$ denote the matching matrix, where $m_{n,k} \in \{0, 1\}$ indicates whether $f_n$ and $\mathcal{P}_k$ are matched or not. The loss defined in Equ. 6 is updated as

$$\mathcal{L}_{dmpt} = \underset{\mathbf{m}}{\arg\min} \ \text{CE}\left( \sum_{n=1}^{N} \sum_{k=1}^{K} m_{n,k} \left( f_n(\mathcal{P}_k) - \alpha \hat{d}_k(x_k, y_k) \right), Y \right), \tag{7}$$

$$s.t. \quad \forall k, \ \sum_{n=1}^{N} m_{n,k} \leq 1, \ \forall n, \ \sum_{k=1}^{K} m_{n,k} \leq 1. \tag{8}$$

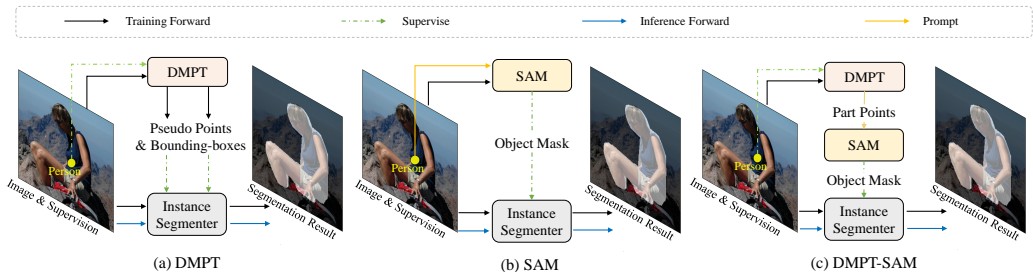

Figure 3: Comparison of single point prompt and DMPT prompt in segment anything model (SAM).

Equ. 8 defines the matching constraint so that a part token is only assigned to a single part classifier, which avoids semantic aliasing among object parts. During training, the optimal matching problem is solved by a weight bipartite matching algorithm (et al, 2001), *e.g.*, the Hungarian algorithm (Kuhn, 1995). Optimized by Equ. 7 with a deformation constraint term $(\hat{d}_k(x_k, y_k))$, DMPT not only learns stable and various fine-grained semantics but also is robust to object deformation.

### 3.4 PART-BASED GUIDANCE GENERATION

Once Equ. 7 is optimized, we obtain the matched part tokens $\mathbf{P}$ to generate guidance information. As previously discussed, to enable the self-attention mechanism to activate fine-grained semantics, we first use part tokens to create a set of part points $\mathbf{G} = \{\mathcal{G}_k, k = 1, 2, ..., K\}$, where $\mathcal{G}_k = \{(x_k, y_k)\}$ denotes the part coordinates as $\mathcal{G}_k = \arg\max_{x_i, y_i}(s_k(x_i, y_i) - \alpha \hat{d}_k(x_i, y_i))$. We then add an MLP layer atop the self-attention network to predict these part points, denoted as $\hat{\mathbf{G}} = \{\hat{\mathcal{G}}_k = (\hat{x}_k, \hat{y}_k), k = 1, 2, \ldots, K\}$, Fig. 2. This point prediction procedure is driven by updating the bipartite matching loss defined in Equ. 1 as

$$\mathcal{L}_{obj} = \mathcal{L}_{loc} + \mathcal{L}_{cls} + \mathcal{L}_{part}(\hat{\mathbf{G}}, \mathbf{G}), \tag{9}$$

where $\mathcal{L}_{part}(\hat{\mathbf{G}}, \mathbf{G}) = \frac{1}{2K}\left(\sum_{i=1}^{K}\min_j ||(x_i, y_i) - (\hat{x}_j, \hat{y}_j)||_2 + \sum_{j=1}^{K}\min_i ||(x_i, y_i) - (\hat{x}_j, \hat{y}_j)||_2\right)$ is the Chamfer loss defined in (Fan et al., 2017). Through gradient backpropagation during training, Equ. 9 drives the self-attention mechanism to captures information about these part points, enabling the model to possess part-level semantic awareness.

### 3.5 POINTLY SUPERVISED INSTANCE SEGMENTATION

**DMPT.** Fig. 3(a) shows the PSIS framework with DMPT. The instance segmenter is implemented upon a Mask R-CNN (He et al., 2017) framework, which consists of a bounding-box detection head and an instance segmentation head. Our DMPT generates a pseudo bounding-box, which encloses the maximum connected object area within the self-attention map, indicating the position of $M^+$. For mask supervision, we regard the pixels which within part tokens and re-estimated ones as foreground. The background pixels is sampled from pixels with small attention value within the pseudo bounding-box. **DMPT-SAM.** Combed with the segment anything model (SAM), DMPT updates the point prompt learning for instance segmentation, Fig. 3(b). Compared with the conventional point prompt, Fig. 3(c), DMPT facilities estimating fine-grained semantics related to object parts, and thereby alleviates the semnatic ambiguity of point prompt learning.

## 4 EXPERIMENT

### 4.1 IMPLEMENTATION DETAILS

We implement DMPT upon imTED (Zhang et al., 2022b). When training DMPT, random horizontal flips and auto-augmentation on multi-scale ranges are used for data augmentation. DMPT is trained with AdamW optimizer with batch size 16 on 8 Tesla A100 GPUs. The weight decay and training epoch are 0.05 and 12 respectively. The learning rate is initialized as 0.0001, and reduced by a magnitude after 9 and 11 epochs. Following BESTIE (Kim et al., 2022), we select the center of ground-truth bounding-boxes as the supervision point to compare with the state-of-the-art approaches.

| Method | Backbone/Params. | Supervision | mAP$_{25}$ | mAP$_{50}$ | mAP$_{75}$ |
|---|---|---|---|---|---|
| Mask R-CNN(ViT) (Zhang et al., 2022a) | ViT-S/22.1M | $\mathcal{M}$ | 77.2 | 68.3 | 46.0 |
| Label-Penet (Ge et al., 2019) | VGG-16/134M | $\mathcal{I}$ | 49.2 | 30.2 | 12.9 |
| CL (Hwang et al., 2021) | ResNet-50/25.6M | $\mathcal{I}$ | 56.6 | 38.1 | 12.3 |
| BESTIE (Kim et al., 2022) | HRNet-W48/63.6M | $\mathcal{I} + \mathcal{W}$ | 53.5 | 41.8 | 24.2 |
| IRNet (Ahn et al., 2019) | ResNet-50/25.6M | $\mathcal{I}$ | - | 46.7 | 23.5 |
| WISE-Net (Laradji et al., 2020) | ResNet-50/25.6M | $\mathcal{P}$ | 53.5 | 43.0 | 25.9 |
| BESTIE (Kim et al., 2022) | HRNet-W48/63.6M | $\mathcal{P} + \mathcal{W}$ | 58.6 | 46.7 | 26.3 |
| Point2Mask (Li et al., 2023) | Swin-L/197M | $\mathcal{P}$ | - | 55.4 | 31.2 |
| Point2Mask (Li et al., 2023) | ResNet-101/44.5M | $\mathcal{P}$ | - | 48.4 | 22.8 |
| AttnShift (Liao et al., 2023) | ViT-S/22.1M | $\mathcal{P}$ | 68.3 | 54.4 | 25.4 |
| DMPT*(ours) | ViT-S/22.1M | $\mathcal{P}$ | 68.6 | 54.5 | 27.4 |
| DMPT(ours) | ViT-S/22.1M | $\mathcal{P}$ | **69.6** | **56.4** | **30.0** |
| DMPT-SAM(ours) | ViT-S/22.1M | $\mathcal{P} + \mathcal{S}$ | **70.7** | **59.4** | **35.5** |
| BESTIE[†] (Kim et al., 2022) | ResNet-50/25.6M | $\mathcal{P}$ | 66.4 | 56.1 | 30.2 |
| AttnShift[†] (Liao et al., 2023) | ViT-S/22.1M | $\mathcal{P}$ | 70.3 | 57.1 | 30.4 |
| DMPT[†](ours) | ViT-S/22.1M | $\mathcal{P}$ | **72.3** | **60.2** | **32.5** |

Table 2: Performance on the PASCAL VOC 2012 $val$ set. $\mathcal{M}$ denotes pixel-wise mask annotations. $\mathcal{I}$ and $\mathcal{P}$ respectively denotes image-level and point-level supervisions. $\mathcal{W}$ indicates weakly supervision segmentation results as supervision and $\mathcal{S}$ prompting SAM with ViT-Base for object mask annotations. * denotes supervised by pseudo-center points and [†] applying retraining. Note that the PSIS performance of Point2Mask is evaluated by its official code.

| Method | Backbone/Params. | Supervision | MS-COCO val2017 | | | MS-COCO test-dev | | |
|---|---|---|---|---|---|---|---|---|
| | | | AP | AP50 | AP75 | AP | AP50 | AP75 |
| Mask R-CNN(ViT) (Zhang et al., 2022a) | ViT-S/22.1M | $\mathcal{M}$ | 38.8 | 61.2 | 41.3 | 38.9 | 61.5 | 41.7 |
| BESTIE (Kim et al., 2022) | HRNet-W48/63.6M | $\mathcal{I}$ | 14.3 | 28.0 | 13.2 | 14.4 | 28.0 | 13.5 |
| LIID (Liu et al., 2022) | ResNet-101/44.5M | $\mathcal{I}$ | - | - | - | 16.0 | 27.1 | 16.5 |
| WISE-Net (Laradji et al., 2020) | ResNet-50/25.6M | $\mathcal{P}$ | 7.8 | 18.2 | 8.8 | - | - | - |
| BESTIE (Kim et al., 2022) | HRNet-W48/63.6M | $\mathcal{P} + \mathcal{W}$ | 17.7 | 34.0 | 16.4 | 17.8 | 34.1 | 16.7 |
| Point2Mask (Li et al., 2023) | Swin-L/197M | $\mathcal{P}$ | 14.6 | 29.5 | 13.0 | - | - | - |
| Point2Mask (Li et al., 2023) | ResNet-101/44.5M | $\mathcal{P}$ | 12.8 | 26.3 | 11.2 | - | - | - |
| AttnShift (Liao et al., 2023) | ViT-S/22.1M | $\mathcal{P}$ | 19.1 | 38.8 | 17.4 | 19.1 | 38.9 | 17.1 |
| DMPT(ours) | ViT-S/22.1M | $\mathcal{P}$ | **20.7** | **41.7** | **19.3** | **20.8** | **41.5** | **19.7** |
| DMPT-SAM(ours) | ViT-S/22.1M | $\mathcal{P} + \mathcal{S}$ | **22.7** | **45.5** | **21.5** | **22.6** | **45.7** | **21.4** |

Table 3: Performance on MS-COCO 2017 $val$ and $test\text{-}dev$ set.

We also report the performance trained with pseudo-center points (Chen et al., 2022) which simulates real annotations. We choose the box-center point as a point prompt for generating object mask. PASCAL VOC 2012 (Mark et al., 2010) and MS-COCO datasets are used for performance evaluation.

## 4.2 PERFORMANCE

In Table 2, DMPT is compared with the state-of-the-art methods on the PASCAL VOC 2012 $val$ set. It outperforms AttnShift (Liao et al., 2023) by a significant margin of 2.0% (56.4% $vs$ 54.4%) upon mAP$_{50}$ metric. For mAP$_{75}$, DMPT achieves 30.0%, 4.4% better than that of AttnShift, demonstrating the superiority of part-based modeling mechanism than the clustering-based method. Combined with SAM, DMPT-SAM achieves 59.4% on mAP$_{50}$, setting the state-of-the-art benchmark for PSIS.

In Table 3, DMPT outperforms AttnShift by 1.6% AP (20.7% $vs$ 19.1%). For the AP50 metric, DMPT surpasses AttnShift by 2.9% (41.7% $vs$ 38.8%) and 1.9% (19.3% $vs$ 17.4%) for AP75. These results demonstrate the DMPT's superiority of handling appearance variances and deformation of objects with an iterative procedure among part token allocation and token-classifier matching to learn various fine-grained semantics and part deformation.

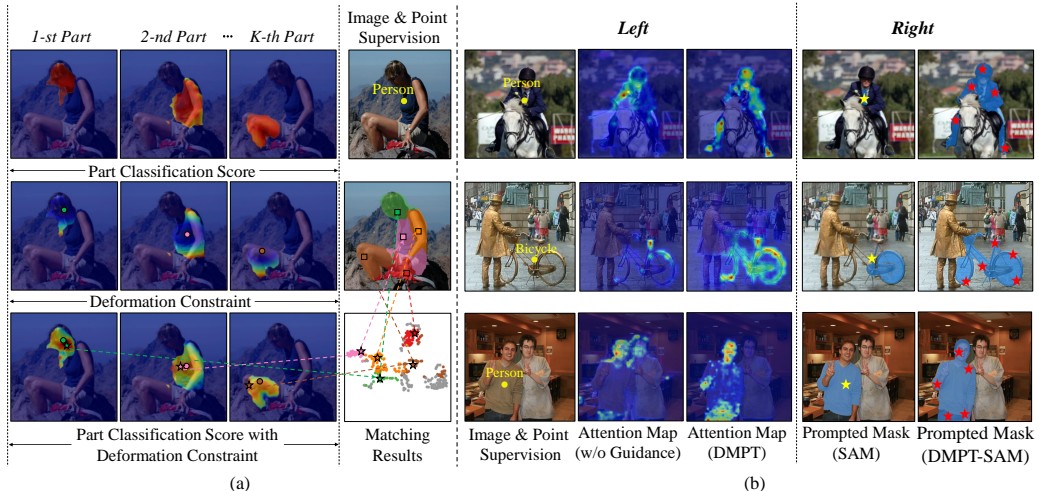

Figure 4: **(a)**: Visualization of part token allocation (heat-maps in 1-3 columns) and token-classifier matching (4 column). From the first to the third rows are part classification score, deformation constraint, and part classification score with deformation constraint, respectively. The allocated part tokens are matched with the part classifiers. **(b)**: Comparison of attention maps (Left: heat-maps in 2-3 columns) and generated masks by prompting SAM (Right: 4-5 columns). (Best viewed in color)

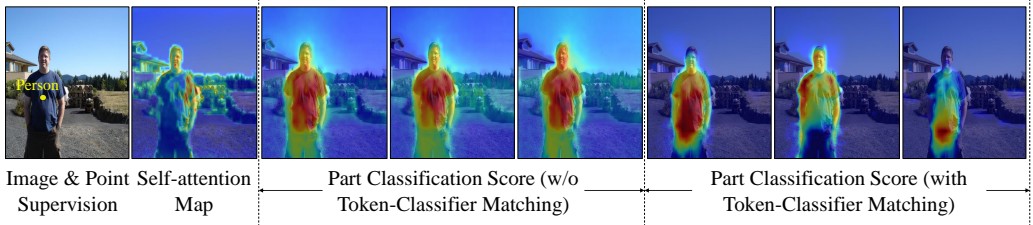

Figure 5: Self-attention map (column 2), activation map of part classifier trained w/o token-classifier matching (columns 3-5), and w/ token-classifier matching (columns 6-8). (Best viewed in color)

## 4.3 VISUALIZATION ANALYSIS

**Part Token Allocation.** Heat-maps in columns 1-3 in Fig 4(a) demonstrate part token allocation by presenting the part classification score, deformation constraint, and part classification score under deformation constraint. The part classification score highlights the part regions but is unable to allocate part tokens. By combining the part classification score with the deformation constraint, the token with a high part classification score while close to the anchor points (colored cycles) is allocated as part tokens (colored rectangles and pentagrams).

**Token-Classifier Matching.** Column 4 of Fig 4(a) shows the token-classifier matching results. Each matched part token represents a cluster, Fig 4(last row). Using these part tokens as supervision, DPMT predicts the precise mask for the whole object extent, Fig 4(the second row). Fig. 5 further shows the effect of token-classifier matching. The self-attention maps (column 2) generated by ViT and part classifiers without token-classifier matching (columns 3-5) falsely activate backgrounds and/or object parts. DMPT discriminatively activates fine-grained semantics as well as suppressing background noise, Fig. 5(columns 6-8).

**Part-based Guidance Generation.** Heat-maps in the left of Fig 4(b) show improvement of attention maps when DMPT performs part-based guidance, where more semantic regions are activated. This validates DMPT enhances the self-attention mechanism towards accurate instance segmentation.

## 4.4 DMPT-SAM

In Table 1, DMPT-SAM respectively improves the SAM on PASCAL VOC 2012 11.3%, 19.5% and 16.5% upon $mAP_{25}$, $mAP_{50}$ and $mAP_{75}$ On MS-COCO 2017, DMPT-SAM achieves 3.1%, 8.9% and 2.6% improvements, showing the potential to reform point prompt learning. Fig. 4(b) in the right shows that DMPT-SAM generates more complete and accurate masks by prompting with part points.

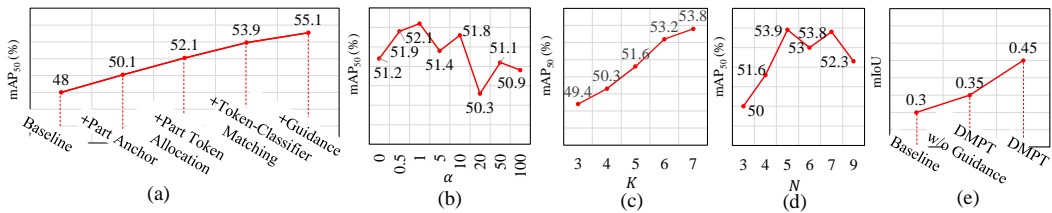

Figure 6: Ablation results. (a) Module ablation. (b) Regularization factor $\alpha$. (c) Number of part tokens $K$. (d) Number of part classifiers $N$. (e) Segmentation accuracy of self-attention map.

## 4.5 ABLATION STUDY

All results in this section are evaluated on the PASCAL VOC 2012 $val$ set.

**Baseline.** Following the previous weakly supervised object detection methods (Liao et al., 2022), we leverage the self-attention maps in ViT to generate pseudo bounding-boxes and use them as the supervision of the detection head. The baseline method randomly samples points in the high-confidence area of self-attention attention map as foreground points and low-confidence area as background ones as the supervision points of the segmentation head. As shown in Fig. 6(a), the baseline achieves 48.0% $mAP_{50}$.

**Part Anchor.** We replace the randomly sampled foreground points with the part "anchor" points which indicate the center of clusters generated by mean-shift method (*i.e.* defined in Equ. 2). As shown in Fig. 6(a), the $mAP_{50}$ is significantly improved by 2.1% (50.1% $vs$ 48.0%), which indicates the token located at "anchor" point has high probability to present an object part.

**Part Token Allocation.** In Fig. 6(a), when using the location of part tokens estimated by the part classifier with deformation constraint as segmentation supervision, the $mAP_{50}$ is further improved by 2.0% (52.1% $vs$ 50.1%), demonstrating the part token owns discriminant ability to accurately present fine-grained semantic. We conduct experiments with different regularization factors $\alpha$ (defined in Equ.5) in Fig.6(b). $\alpha = 5.0$ reports the best $mAP_{50}$ (52.1%). When $\alpha = 0$, only part classification score determines whether a patch token should be allocated as the part token, which reduces the $mAP_{50}$ by 0.9% (51.2% $vs$ 52.1%). This indicates that introducing deformation constraint to classifier learning can enhance the discriminate ability to locate object with deformation.

**Token-Classifier Matching.** In Fig. 6(a), token-classifier matching brings 1.8% (53.9% $vs$ 52.1%) gain upon $mAP_{50}$, which implies this mechanism optimally matches part tokens and with classifiers. When fixing the number of part classifiers ($N = 7$), we analyze the number of part token $K$. In Fig. 6(c), DMPT achieves 53.8% when $K = 7$. Mismatched number of tokens ($K = 3$) and classifiers ($N = 7$) could result in insufficient optimization on stable fine-grained semantics. We further conduct experiments with respect to the number of part classifier $N$ in Fig. 6(d), where we set $K = N$. It shows that the best performance 53.9% is achieved with $N = 5$. Insufficient ($N = 3$) parts cause less accurate part representation (50.6%). More parts ($N = 9$) could result in over-segmenting the objects and reduces the performance to 52.3%.

**Part-based Token Guidance Generation.** With part-based guidance, DMPT further gains 1.2% (55.1% $vs$ 53.9%) improvement, Fig. 6(a). In Fig. 6(e), we calculate mIoU between self-attention maps and ground-truth masks to evaluate the instance segmentation ability. With part-based guidance, DMPT achieves 0.45 mIoU, 0.1 higher than that without guidance, demonstrating DMPT promotes self-attention mechanism to capture fine-grained semantics and activate towards full extent of objects.

## 5 CONCLUSION

We proposed discriminatively matched part tokens (DMPT) to handle object deformation and appearance variances in pointly supervised instance segmentation. DMPT incorperated part classifiers and matches them with part tokens to improve token discrimination capacity and avoid semantic aliasing. Extensive experiments validated that DMPT set a new state-of-the-art performance for pointly supervised instance segmentation using ViT, as well as demonstrating the potential to reform point prompt learning.

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
