# OpenReview forum: "Discriminatively Matched Part Tokens for Pointly Supervised Instance Segmentation"
_ICLR.cc/2024/Conference — Submitted to ICLR 2024_

### Official Review · Reviewer_Br6s · 2023-10-30

**Soundness:** 3 good
**Presentation:** 3 good
**Contribution:** 3 good
**Rating:** 6
**Confidence:** 4

**Summary:**

This paper aims to achieve pointly supervised instance segmentation based on self-attention and propose discriminatively matched part tokens (DMPT) method to address the deformation and variations in appearance of object.  This method first allocates a token for each object part by finding a semantic extreme point, and then prensents part classifiers with deformable constraint to re-estimate part tokens which are utilized to guide and enhance the fine-grained localization capability.  The extensive experiments are conducted and show the effectiveness of the method.
Besides, this method can enhance SAM model to achieve the better performance.

**Strengths:**

1. The proposed DMPT sounds reasonable based on self-attention.

2. The performance  on PSIS is state-of-the-art compared with the current methods. And the method can benefit the performance of SAM model for object-level segementation.

3. This paper is well-conducted, including the presentation  and figures, tables.

4. The experimental section is well-presented to demonstrate the effectivenss.

**Weaknesses:**

1. The section of related work is inadequate in weakly supervised instance segmentation.  Some weakly image-level supervised methods[1] and box-supervised methods[2][3]  not listed.
2.  The inference speed of whole method could be reported and it can better demonstrate the superiority of the proposed method.
3.  Some typos  exist, like "Combed with ..." in  section 3.5 should be "Combined with ..."

[1] Ahn et al. Weakly supervised learning of instance segmentation with inter-pixel relations, CVPR2019.
[2] Tian et al. Boxinst: High-performance instance segmentation with box annotations, CVPR2021.
[3] Li et al. Box-supervised instance segmentation with level set evolution, ECCV2022.

**Questions:**

1. If some parts of an object are directly filtered as background due to low activation values in the self-attention map, how could this be further optimized in subsequent steps?
2. It would be beneficial to include comparsions with more weakly supervised methods as the performance reference for the readers.

---

> ### Author Response · Authors · 2023-11-19
> **Rebuttal by Authors**
>
> ## Response to Weakness
> W1: The section of related work is inadequate in weakly supervised instance segmentation. Some weakly image-level supervised methods[1] and box-supervised methods[2][3] not listed.
>
> R1:As suggested, we have integrated the previous researches, including IRNet, BoxInst and Box-LevelSet, into the section of related work for a comprehensive survey.
>
> W2: The inference speed of whole method could be reported and it can better demonstrate the superiority of the proposed method.
>
> R2: In the supplementary, we measured and reported the FPS metric of our DMPT model and others, which displays DMPT's superiority of inference efficiency.  As suggested, we will move this section in the main paper to improve the presentation of our strengths.
>
>
> W3: Some typos exist, like ''Combed with ..." in section 3.5 should be ''Combined with ..."
>
> R3: Thinks for your advice, the typos have been revised and we think the fluency of our paper has been enhanced.
>
> ## Answer to Questions
> Q1: If some parts of an object are directly filtered as background due to low activation values in the self-attention map, how could this be further optimized in subsequent steps?
>
> A1: While some object parts are occluded in some images, in statistical sense, all object parts of the same category will appear in the dataset. Specially, our part classifiers statistically learn the fine-grained semantics from the whole dataset. As long as most of the parts visible in the images, part classifiers can be optimized to represent and recognize the stable fine-grained semantics by using visible parts. In addition, our token-classifier matching mechanism allows the part classifier not matching any parts if its corresponding part is occluded. This characteristic ensures that the classifiers not only are able to represent the fine-grained pattern from visible parts, but also can keep steady optimized when object part absents in some images.
>
> Q2: It would be beneficial to include comparisons with more weakly supervised methods as the performance reference for the readers.
>
> A2: Thanks for your constructive and insightful comments. We have added more comparisons with weakly supervised methods in the revised version.

---

### Official Review · Reviewer_bde7 · 2023-11-01

**Soundness:** 3 good
**Presentation:** 1 poor
**Contribution:** 2 fair
**Rating:** 6
**Confidence:** 4

**Summary:**

This paper introduces Discriminatively Matched Part Tokens (DMPT) to extend the capabilities of self-attention in point-based supervised instance segmentation. The main working logic of DMPT is as follows: 1) perform mean-shift to find part tokens, 2) update the part tokens based on part deformation constraint, and 3) match the part tokens with the part classifiers. Through iterative optimization, DMPT identifies the most discriminative part tokens, enabling the capture of fine-grained semantics and activation of the complete object extent. Extensive ablation studies and comparisons with the other methods are conducted on the PASCAL VOC and MS-COCO datasets. Notably, DMPT also can be integrated with the Segment Anything Model (SAM).

**Strengths:**

1. The idea is intuitive. It can be a simple yet effective approach.
2. The performance seems pretty good for both with and without using SAM.
3. Extensive ablation studies are conducted.

**Weaknesses:**

[Major]

1. The authors present the result for an image of a person. It would be advantageous to include more image samples in the main paper. I am particularly interested in the extent to which the part-classifiers effectively learn semantically meaningful parts and consistently activate similar parts in diverse images. Interestingly, the person sample in Figure 2 in the supplementary material does not seem to achieve this. Could the authors explain this?

2. I have reservations about the validity of the token-classifier matching, especially in the following two scenarios. In the rebuttal, visual results for these cases would be appreciated:

* When some parts are missing in the input image due to occlusion or other factors. In such situations, do the part-classifiers corresponding to the missing parts get correctly excluded in the matching matrix?

* Additionally, does the matching mechanism adequately handle cases of over-segmentation? It seems possible that sometimes K can significantly exceed N, especially as there is no constraint on K. In such cases, a single part-classifier should ideally be matched with multiple tokens.

3. It would be valuable for the authors to explain their criteria for determining N, the number of part-classifiers. The optimal number of parts may vary across different classes and datasets. Complex classes like bicycles might require more parts, while classes with simple shapes (e.g., a ball) may need fewer. Can the authors elaborate on their approach to determining the number of part classifiers in various scenarios?

4. I think SAM itself can handle this task to some extent. Using the given point annotations serve as prompts for SAM, we can obtain pseudo-labels simply. Can you check this setting and compare it with your DMPT-SAM?

[Minor]

The notations are somewhat distracting, but honestly, I haven't come up with a better alternative. The core concept of this paper appears to be pretty intuitive to me. However, its mathematical formulation makes the understanding rather complex. It would greatly enhance the paper's clarity if the authors could improve its presentation.

**Questions:**

Please refer to the weaknesses.

---

> ### Author Response · Authors · 2023-11-19
> **Rebuttal by Authors**
>
> R1: Due to challenge of less supervision information, the location precision of object remains imperfect in pointly supervised instance segmentation, which limits accuracy of part tokens generation. Meanwhile, part deformation and occlusion result in difficulty for clustering methods to produce part tokens to represent anthropogenic defined object parts, like "head" or "leg". This does limit the ability of the part classifier to actually activate the object part uniformly.
>
> Surprisingly, DMPT is still able to learn relative stable and consistent fine-grained semantics than the baseline method.
> We conduct statistical analysis of feature similarity on different aspects in Figure 1, updated in "rebuttal_visualization.pdf" of the new supplementary materials.
> In Figure 1(a), the similarity between part features and background of DMPT is lower than that of AttentionShift, which showcases that our part-based modeling mechanism can effectively suppress background noise, obtaining stable semantics. As we utilize part classifiers to uniformly represent fine-grained semantics, DMPT exhibits stronger discrimination among objects of different categories, In Figure 1(b). Particularly, we calculate the feature similarity of different parts of the same object, Figure 1(c). It can be seen that the part features of DMPT are more discriminative than AttentionShift, implying that we do learn consistent fine-grained semantics.
>
> R2.1: In Equ.8 of main paper, the token-classifier matching allows the case when part tokens' number is less than that of part classifiers ($K \textless N$), due to part occlusion or other factors. We visualize the matching matrix of some cases in Figure 2 of the "rebuttal_visualization.pdf".
>
> R2.2: No, the matching mechanism do not handle cases of over-segmentation, because it has been handled before token-classifier matching. First, we pre-define that each object has $N$ parts, based on which we initialize $N$ part classifiers and partition the object into $N$ cluster centers. Then, similar cluster centers are merged according to their cosine distances, avoiding over-segmentation on consecutive part semantics. After this procedure, we produce $K$ part clusters for each object.  Note that $K$ might be different values for different objects. In order to uniformly represent the number of part clusters across all objects, we use $K$ to control the upper limit of the number of object parts after merging similar clusters. In addition, $K\leq N$ is an implicit setting in our paper.
>
> Thanks for this constructive advice. For more clear presentation, we will add the details about how to handle the over-segmentation problem and relation of $K$ and $N$ in the paper.
>
> R3: $N$ is a hyper-parameter to pre-define the number of classifiers optimized. $N$ is experimentally determined for achieving the balanced performance of all the classes within a dataset.
>
> For special usage for other scenarios, $N$ is empirically set to a relative high value to ensure classifiers can sufficiently cover various fine-grained semantics. As our DMPT owns two characteristics, (i) token-classifier matching allows some classifiers not matching any part tokens during model training (R2.1), and (ii) the number ($K$) of part tokens of different objects dynamically adapts to object semantics (R2.2), part classifiers have the potential to be dynamically optimized with objects with various part semantics. However, mismatched number of tokens ($K=3$) and classifiers ($N=7$) could result in insufficient optimization on stable fine-grained semantics, Figure 6(c) of main paper.
>
> R4: The setting you mentioned is right. However, getting one accurate pseudo-mask needs a selection upon three pseudo-masks when using one single point to prompt SAM. Note that producing three masks from one single point prompt is a design of SAM to address ambiguity of the prompt. In the setting of pointly supervised methods, there is no ground-truth mask as guidance to select the most accurate one. We only can select pseudo-masks leveraging location scores (IoU predictions) of these masks predicted by SAM. However, these scores are usually unreliable for obtaining the most accurate mask.
>
> In order to address this issue, DMPT acts a prompter before prompting SAM by generating part points with the the single point, which we termed as DMPT-SAM. The generated part points contains more spatial information ($e.g.$, the deformation, scale and extent of object), which is less ambiguous than one single point. With these points as prompt, the location scores are more reliable to select the accurate pseudo-masks.
>
> Quantitative results in Table 1 and qualitative visualizations in Figure 4(b) of main paper demonstrate that DMPT-SAM facilitates generating more precise pseudo-labels for pointly supervised instance segmentation.
>
> R5: Thanks for your constructive advice and we will further simplify and refine our notations for higher presentation in the camera-ready version.

---

> ### Comment · Reviewer_bde7 · 2023-11-21
>
> All my concerns have been thoroughly addressed, except for the query regarding SAM.
> I still believe that a comparison between DMPT-SAM and the direct application of SAM would enhance the quality of this paper.
>
> In their response, the authors mentioned SAM's ambiguity, potentially compromising the quality of self-supervision.
> However, I posit that this issue may persist even when employing more prompts through the proposed part-based approaches.
> I still cannot understand why the authors do not address this concern in the most clear and powerful way: quantitative comparison.
> The experiments using SAM directly seem not that difficult to conduct, at least for me.
>
> Nevertheless, I deeply appreciate the authors' extensive efforts for the rebuttal.
> It remains evident that DMPT (excluding SAM) outperforms existing methods, and the overall quality of this paper is pretty good even without DMPT-SAM.
> As a result, I am inclined to raise my rating.

---

> > ### Author Response · Authors · 2023-11-22
> > **Response to Reviewer bde7**
> >
> > Thanks for your support of our work. Your comments inspire us further to improve the combination of DMPT and SAM in the further.

---

### Official Review · Reviewer_QnBw · 2023-11-01

**Soundness:** 3 good
**Presentation:** 3 good
**Contribution:** 3 good
**Rating:** 6
**Confidence:** 4

**Summary:**

The paper presents discriminatively matched part tokens to improve pointly supervised instance segmentation. The part tokens are initialized by clustering and refined by part classifiers. The part tokens are utilized with self-attention maps to generate better pseudo masks for training instance segmentation models. The proposed method is validated on PASCAL 2012 and COCO datasets. The experimental results show that the proposed method achieves state-of-the-art performance for pointly-supervised instance segmentation.

**Strengths:**

The proposed methods utilizes part tokens to generate pseudo masks of higher quality for training instance segmentation masks. Part-classifier matching, spatial constraints and part-based guidance are proposed to generate better part tokens. The design of the components of the proposed method is well motivated.

The proposed method achieves state-of-the-art performance for pointly-supervised instance segmentation. Extensive experiments are conducted to validate the effectiveness of the components of the proposed method. Visualization results show that the proposed method can generate better attention maps for pseudo mask generation.

The proposed method is well written. The idea is clearly presented.

**Weaknesses:**

The training process seems complex. How much computational cost is inctroduced by the newly introduced modules?

It is not clear how much performance improvment is brought by the introduction of spatial constrains (eq. (5)). One more experiment is needed to verify this.

**Questions:**

See Weakness section.

---

> ### Author Response · Authors · 2023-11-19
> **Rebuttal by Authors**
>
> W1: The training process seems complex. How much computational cost is introduced by the newly introduced modules?
>
> R1: We compare the efficiency of training and inference of our method upon the AttentionShift in the following Table.
>
> |  Method   | Training Time  |  FPS |
> |  ----  | ----  |  ---- |
> | AttentionShift  | 9.34h | 14.3 |
> | DMPT  | 9.7h(+0.36h) | 14.3 |
>
> The additional training time (0.36h) spends on the proposed part token allocation and token-classifier matching. Note that the batch size is 16 on 8 Tesla-A100 GPUs, training PASCAL VOC for 36 epochs. And we test the FPS with an image, whose the shortest side is 800 pixels, on a single Tesla-A100.
>
> W2: It is not clear how much performance improvement is brought by the introduction of spatial constrains (eq. (5)). One more experiment is needed to verify this.
>
> R2: As suggested, we have conducted experiment to display the performance of spatial constrains in Figure 6(b) of main paper and analyzed the effectiveness in paragraph "**Token-Classifier Matching**" of Sec 4.5 in the paper. For more clarity, we have revised part of paragraph as following:
>
> ''When $\alpha=0$, only part classification score determines whether a patch token should be allocated as the part token or not, which achieves 51.2\% upon mAP50, 1.1\% higher than that (50.1\% in Fig.6(a)) allocating cluster centers as part tokens, defined in Equ. 2. After introducing part deformation constraint, $\alpha=5$ reports the best mAP$_{50}$ (52.1\%), which outperforms the best method by 0.9\% (52.1\% vs 51.2\%). ''

---

### Official Review · Reviewer_5abY · 2023-11-01

**Soundness:** 3 good
**Presentation:** 3 good
**Contribution:** 3 good
**Rating:** 6
**Confidence:** 3

**Summary:**

The paper proposes a method called discriminatively matched part tokens (DMPT) for pointly supervised instance segmentation. The DMPT method allocates tokens for parts by using the attention maps from the vision transformer and matches the part tokens with part classifiers. In addition, DMPT can generate part points and be combined with SAM for better performance.

**Strengths:**

+ The proposed part token allocation and token classification with deformation constraint are reasonable and effective. They help to recognize more stable object deformation parts.
+ The proposed method is well illustrated with the visualization figures.
+ The corresponding source code is attached to this submission, which reflects good reproducibility.

**Weaknesses:**

- The paper shares the same idea of using self-attention maps with prior works [a], but the differences are not well elaborated.
- The limitation of the part deformation constraint is not considered. For example, will it work properly if the target is a snake with a long and thin shape?

[a] AttentionShift: Iteratively Estimated Part-based Attention Map for Pointly Supervised Instance Segmentation, CVPR'23.

**Questions:**

Please refer the Weaknesses.

---

> ### Author Response · Authors · 2023-11-20
> **Rebuttal by Authors**
>
> W1: The paper shares the same idea of using self-attention maps with prior works (AttnShift), but the differences are not well elaborated.
>
> R1: As our baseline method, AttentionShift generates multiple anchor points to represent object parts. The method roots on parameter-free clustering, like mean-shift, which however experiences difficulty to uniformly represent part semantics and suffers from background noises. As a significant difference, DMPT leverages instance classification loss to train part classifiers so that it can dynamically match part tokens. After the loss gradient back-propagation, the classifier parameters are updated to recognize and represent fine-grained features as well as suppress background noises.
>
> Furthermore, the key-point shift procedure in AttentionShift does not consider part spatial deformation, which might cause key-points false shift to the background or another instances. Convincingly, DMPT leverages learnable parameters and deformation constraint to estimate the deformation of each part token, which facilities handling spatial deformation.
>
> With above essential promotions, DMPT respectively achieves significant (2.0\% mAP$_{50}$ and 1.6\% AP) performance gains on PASCAL VOC 2012 and MS-COCO 2017.
>
> W2: The limitation of the part deformation constraint is not considered. For example, will it work properly if the target is a snake with a long and thin shape?
>
>
> R2: The proposed part deformation constraint has the potential to handle these extreme cases. Specially, thanks to the ability of global attention mechanism to effectively capture the overall deformation of the object, DMPT can dynamically decompose the object into various parts, acting as reference points for local spatial deformation, which makes the model focus attention on object parts. Based upon these reference points, the proposed part deformation constraint further enhances local features by coupling the deformation representation to counter part deformation. Therefore, incorporating our part deformation constraint with the global attention mechanism is crucial for effectively resolving substantial object deformations.
>
> Undoubtedly, there are some limitations of the spatial deformation. On the one head, initializing deformation constrain $d(x_i,y_i)$, defined in Equ.3, as a square distance feature might be less rational, as the part of some objects like snake sometimes represents a slender shape, whose spatial feature should have different initialized weights upon horizontally and vertically. On the other head, the value of $d(x_i,y_i)$ should be dynamically reduced before coupling with part classification scores when meeting extreme cases, as spatial constrain might be less unreliable in this situation.
>
> Thanks for your advice, which inspires us to further improve our method.

---

> > ### Comment · Reviewer_5abY · 2023-11-22
> >
> > Thanks for the authors’ efforts in rebuttal.
> >
> > The authors clarify most of my concerns regarding novelty and limitations of the proposed DMPT. I hope the differences from [a] could be further highlighted in the revised version.
> >
> > After carefully reading the response and other reviewers' comments, I would like to keep my rating.

---

> > > ### Author Response · Authors · 2023-11-22
> > > **Response to Reviewer 5abY**
> > >
> > > Thanks for your support of our work. We will further enhance the presentation upon the difference between our DMPT and AttentionShift in the revised version.

---

### Meta-Review · Area_Chair_h1Nv · 2023-12-06

**Metareview:**

The paper presents a method for point-supervised instance segmentation (called DMPT). DMPT proceeds in a few steps: 1) cluster the self-attention map (using mean-shift) to find *part tokens*. 2) refine the part-tokens using deformation constrains. 3) match the part tokens with part classifiers. Each part token is associated with a part-instance mask, thus yielding a full instance segmentation.
The paper was reviewed by four experts in the field, all of which rated it as borderline.
+ The paper shows promising results
+ Sufficient ablations
- The method has plenty of moving parts, many of which are not very well motivated or contrasted. Clearly the proposed method is a solution, but the authors do not make a very convincing case that it is the right solution. Many reviewers hint at this: Reviewer 5abY - W1,W2, Reviewer QnBw weaknesses, Reviewer bde7 W2, W3. This is in part a presentation issue, and in part an issue with the method itself. While the authors answer most of the reviewers direct requests, the overall problem remains.
- The AC agrees with Reviewer bde7 that a direct comparison with SAM would have made the paper stronger. It is concerning that the authors build on SAM, but even after a request refuse to compare to it directly.

As the initial reviews reflect, at the onset of the review process this paper was a borderline submission. After the discussion, nobody was willing to strongly (or weakly) argue for acceptance. In addition, not all concerns of the reviewers were addressed (see comparison to SAM above). Considering all of this, the paper falls short of the bar for an ICLR publication.

**Justification For Why Not Higher Score:**

This is a true borderline paper. Nobody is willing to argue for acceptance, nobody wants to outright reject it. In the end, the refusal to compare directly to SAM (in whatever form) introduces a risk the AC is not willing to take. The paper would be stronger if resubmitted with all the proper baseline comparisons.

**Justification For Why Not Lower Score:**

N/A

---

### Decision · Program_Chairs · 2024-01-16

Reject